# A Peridynamic Computational Scheme for Thermoelectric Fields

**DOI:** 10.3390/ma13112546

**Published:** 2020-06-03

**Authors:** Migbar Assefa Zeleke, Xin Lai, Lisheng Liu

**Affiliations:** 1Department of Mechanical Engineering, University of Botswana, 0061 Gaborone, Botswana; zelekem@ub.ac.bw; 2Department of Mechanical Engineering, Hawassa University, 05 Hawassa, Ethiopia; 3Department of Engineering Structure and Mechanics, Wuhan University of Technology, Wuhan 430070, China; laixin@whut.edu.cn; 4State Key Laboratory of Materials Synthesis and Mechanics, Wuhan University of Technology, Wuhan 430070, China

**Keywords:** thermoelectricity, heat conduction, electric conduction, peridynamic theory, insulated crack

## Abstract

Thermoelectric materials are materials that involve the coexistence of heat flux and electric current in the absence of magnetic field. In such materials, there is a coupling among electric potential and temperature gradients, causing the thermoelectric effects of Seebeck and Peltier. Those coupling effects make the design and analysis of thermoelectric materials complicated and sophisticated. The main aim of this work is dealing with thermoelectric materials with discontinuities. Since heat and electric fluxes are undefined at the crack tip and the temperature and electric fields across the crack surface are discontinuous, it is better to apply peridynamic (PD) theory to capture such details at the crack tips. Hence, we propose in this paper a PD theory which is suitable in tackling such discontinuities in thermal and electric fields. In this study, the continuum-based electrical potentials and temperature fields are written in the form of nonlocal integrals of the electrical potentials and temperature that are effective whether we have discontinuities or not. To illustrate the consistency of the peridynamic technique, a number of examples were presented and witnessed that PD results were in good agreement with those results from the literature, finite element solutions and analytical solutions.

## 1. Introduction

Energy plays a prominent role in our daily lives. We use energy for heating, cooling, lighting, cooking, and traveling from place to place. Due to rapid population growth and industrialization, the global energy demand increases in an alarming fashion. Accordingly, the International Energy Outlook (IEO) (IEO2016) [1] predicted that there will be a dramatic increase in the world’s energy consumption over the coming 30 years. As we all know, energy from liquid fuels emits more carbon to the environment since they are hydrocarbon fuels. According to IEO2016 prediction, by 2040 43.2 billion metric tons of CO_2_ will be emitted to the environment with an increase of 34% from 2012. Apart from CO_2_ emission there is a huge amount of heat loss through coolant liquid, hot exhaust gas, conduction, convection and radiation from the transportation and industrial sectors. Therefore, it is quite critical to find renewable energy sources that satisfy the global energy demand while reducing CO_2_ emission. 

In general, about 70% of the energy generated by gasoline engine is wasted in the form of heat [2]. Hence, it is important to devise a technique to extract clean energy from the waste heat using thermoelectric devices. Thermoelectric convertors are solid-state devices that can convert waste heat into electricity and vice versa. These devices are reliable due to the fact that they do not have any moving parts. Therefore the energy generated from the waste heat using thermoelectric convertors not only improves the efficiency of energy production but also reduces CO_2_ emission, and hence it is clean and reliable [3].

In practice, thermoelectric convertors are generally engaged in places with an extremely high temperature difference, which leads to an expansion and contraction of the device periodically. Such cyclic thermal load creates cracks within the material that considerably affect the service life of the convertor, which in turn exposes the device to high thermal stresses and strains. Therefore, it is quite important to design thermoelectric convertors to tolerate a huge amount of thermal cyclic load that can possibly result in mechanical fatigue [4]. Hence, the life cycle prediction of thermoelectric convertors is quite central from the design viewpoint [5]. 

Furthermore, the analysis of coupled fields is one of the challenging aspects in the design of thermoelectric convertors using classical theories. In a coupled thermoelectric phenomenon, there exists the coupling of heat and electricity. The equations for a coupled thermoelectric system based on the conventional continuum mechanics are well developed [6,7]. For homogeneous and isotropic thermoelectric materials, the thermoelectric constitutive equations considered the thermal flux, which is related to the gradient of temperature, and the charge flux, which is related to electric potential gradient. Seebeck in 1821 and Peltier in 1835 observed the thermoelectric effect well before the quantitative formulation of Ohm’s law in 1855.

Due to imperfection in the manufacturing process of thermoelectric materials (TEMs), flaws like microcracks, scratches and other discontinuities are unavoidable. As a result of these discontinuities, there will be thermal flux, stress, and electric current concentration near the discontinuities and results in an interruption of operation or failure. Therefore, it is quite imperative to give considerable attention to the analysis of thermoelectric materials with discontinuities. 

The modeling and analysis of thermoelectric materials (TEMs) with cracks or defects using the well-known classical theories results in an infinite flux at the discontinuities. These fluxes are infinite at discontinuities due to the fact that classical theories are formulated on the basis of the partial differential equations (PDEs) that govern the physical phenomena. One of the widely used and multipurpose numerical techniques that solve engineering problems is Finite Element Method (FEM), which is formulated based on PDEs. This method has been used for more than seven decades in solving a huge number of engineering problems with good accuracy. Although FEM is quite successful and popular in solving engineering problems, it has its own weaknesses in the modeling and analysis of discontinuities. To circumvent the limitations encountered by classical theories, Stewart Silling [8] developed a new and fascinating theory called peridynamic (PD) theory. For the last couple of decades, this new theory significantly attracted the attention of researchers in the area of science and engineering due to its potential in solving both continuous bodies and bodies with discontinuities. Mathematically speaking, classical theories are formulated based on partial differential equations that are undefined at features like voids, cracks, and other discontinuities. Hence the classical formulation must be supplemented with additional criteria to properly represent and predict the behavior of discontinuities [9,10,11,12]. PD, on the other hand, does not demand any other criterion to represent and predict the behavior of discontinuities. Therefore, this article is aimed to develop a peridynamic formulation to study the thermal and electrical transport processes of thermoelectric materials.

## 2. Peridynamics (PD) Formulation

PD theory is an extension of a nonlocal continuum model without any spatial derivatives [13]. This nonlocal continuum formulation was originally formulated to deal with fracture and elastic deformation in solids [8]. In this article [8], the formulation is limited to linear, isotropic, and elastic materials with effective Poisson’s ratio in 3D and 2D plane strain is ¼, but for 2D plane stress it is 1/3. To circumvent the aforementioned restrictions, Silling et al. [14] proposed a state-based PD technique where the bond force depends on the collective deformation of the participating bonds connected with the material points within a given domain called the horizon. Peridynamic theory has been used for the last several years to model fracture and elastic deformation in solids [8,14,15,16]. Though the literature in the area of peridynamic theory is enormous, we will review only those articles that are pertinent to couple fields and diffusion in PD framework [17,18,19,20,21,22,23,24,25,26,27,28,29,30,31,32,33,34,35,36,37,38,39,40,41,42,43]. Gerstle et al. [20] were the first to propose the analytical and computational simulation of electromigration that accounts for heat transfer in a one-dimensional problem. Later, Bobaru and Duangpanya [26,27] introduced the bond-based PD formulation for thermal problems with evolving discontinuities. Recently, the generalized state-based PD heat-transfer problem by the Lagrangian formulation was demonstrated by Oterkus et al. [28]. In this work [28], the authors determined the PD material parameter, the microconductivity, by simplifying the state-based PD heat-transfer equation to the bond-based PD heat transfer equation, and it was confirmed that the governing equation represented the conservation of thermal energy. Chen and Bobaru [29] analyzed the behavior of PD solutions for transient heat diffusion model and studied the convergence properties of the one-point Gauss quadrature scheme. Oterkus et al. [30] utilized a fully coupled peridynamic formulation to predict the effect of porous flow on deformation fields simultaneously. The authors further investigated the initiation and propagation of fracture using a fully coupled poroelastic PD formulation. Recently, L. Wang et al. [31] derived a state-based heat conduction peridynamic equation by taking into account the effect of non-Fourier and nonlocal phenomena simultaneously. The authors introduced the notion of dual phase lags (DPL) into the framework peridynamic theory. M Dorduncu [32] on the other hand applied peridynamic differential operator (PDDO) to study the steady-state heat conduction in plates with and without insulated cracks. More contributions from J. Zhao et al. [33] introduced the transient advection–diffusion PD model by utilizing the approach developed by Bobaru and Duangpanya [27]. 

Some other applications of PD for thermomechanical and electromechanical coupling can be found in [[23],[34],[35],[36],[37],[38],[39],[40],[41],[42],]. Chen et al. [25] applied implicit PD formulation in the framework of MOOSE to study coupled thermomechanical problems. In this article, the authors used bond-based PD (BB–PD) formulation with regular square discretization. Later the same authors [41] extended their study by reformulating the classical bond-based PD (BB–PD) formulation to solve thermomechanical problems using irregular domain discretization. A more general and interesting formulation of ordinary state-based PD formulation to solve thermomechanical problems with irregular non-uniform domain discretization can be found in [42]. Zhang and Qiao [34] presented a new PD Scheme by extending the ordinary state-based peridynamic (OSB–PD) model to simulate damage initiation and propagation of bimaterials under thermomechanical loading. Wang et al. [35], on the other hand, investigated a coupled thermomechanical BB-PD to simulate thermal shock cracking in rocks. The authors in this article also investigated the effect of inhomogeneous properties and the coefficients of thermal expansion on the cracking patterns. Very recently, Bazazzadeh et al. [36] effectively developed a thermomechanical PD model by exploiting the advantages of adaptive grid refinement to solve the crack propagation problem in ceramic materials. Wildman and Gazonas [23] applied a PD approach to study the failure of dielectric solids that are subjected to electric fields. In the peridynamic perspective of coupled electromechanical and electrical conduction models, Prakash and Seidel [37] explored the beauty of PD to investigate the piezoresistive and electrical response of composite materials at nano scale by introducing electron hopping. Further Prakash and Seidel [38,39] employed a coupled electromechanical PD framework to model the damage- and deformation-sensing capabilities of explosive materials without considering electron hopping. A recent work of Diana and Carvelli [40] implemented micropolar PD (MPPD) formulation to solve electromechanical problems by coupling the electrical conduction PD model with the mechanical micropolar formulation which removed Poisson’s ratio restrictions.

To the best of our knowledge, there are only a few attempts [24,43] in the analysis of both the thermal and electric fields in thermoelectric plate with discontinuities using PD theory. Hence, the main aim of this study is to solve two-dimensional heat and electric conduction problem of thermoelectric plate with discontinuities by applying the PD approach to show the practicality of the proposed formulation. Before we start our discussion on the coupled thermoelectric phenomena, we treat both heat and electric conduction processes distinctly. Here, we first deliberate on the PD formulation of electrical conduction, followed by PD formulation of thermal conduction, and finally the coupled thermoelectric phenomena.

### 2.1. PD Electric Conduction 

In electrical conduction phenomena, material points exchange electrical flux with material points inside its integration domain defined by the horizon $H_{x}$ as shown in Figure 1. With reference to Figure 1, consider the electrical conduction bond $\xi =\left|X^{B}-X^{A}\right|$ that connecs material points $X^{A}$ and $X^{B}$ having volumes ${\mathrm{dV}}_{X^{A}}$ and ${\mathrm{dV}}_{X^{B}}$, respectively. The conductivity of electrical conduction bond ξ is designated by ${\;k}_{b}\left[X^{A},X^{B}\right]$. At this point we assume electric current flows only along the length of the electrical conduction bond, material points interact in a pair-wise manner, and the bonds have an electrical resistance of zero.

Based on the assumptions above, we can express the volumetric electrical flux of current passing through the bond in terms of the potential difference based on Ohm’s law as follows:(1)$$j_{b}\left[X^{A},X^{B}]=-{\;k}_{b}[X^{A},X^{B}\right]\left(\frac{\Phi \left[X^{B}\right]-\Phi \left[X^{A}\right]}{\left|\xi \right|}\right)e\left[X^{A},X^{B}\right]$$
where $e\left[X^{A},X^{B}\right]=\frac{\xi }{\left|\xi \right|}$, $e\left[X^{A},X^{B}\right]$ is the unit vector along the bond, $j_{b}$ is volumetric electrical flux, Φ is Electric Potential, $\left|\xi \right|$ is the magnitude of the bond vector, and ${\;k}_{b}\left[X^{A},X^{B}\right]\;$ is conductivity of the electrical bond. Equation (1) represents the PD current flowing through a single electrical conduction bond. Using the balance of charge for the bond, the time rate of change of total charge in the bond per unit bond volume must be equal to the net current flow through the bond per unit bond volume. In the presence of charge source term at material point $X^{A}$, the PD electrical conduction equation is given by
(2)$$\frac{\partial \rho_{\Omega }\left[X^{A},X^{B}\right]}{\partial t}={\;k}_{b}\left[X^{A},X^{B}\right]\left(\frac{\Phi \left[X^{B}\right]-\Phi \left[X^{A}\right]}{{\left|\xi \right|}^{2}}\right)+{\;J}_{b}\left[X^{A},X^{B}\right]$$
where $\rho_{\Omega }$ is total charge per unit volume. 

To obtain the conservation of charge equation for material point $X^{A}$, due to current flow in all the bonds attached to point $X^{A}$ in its horizon ${\;H}_{X^{A}}$, we integrate Equation (2) over the horizon of $X^{A}$.
(3)$$\int_{{\;H}_{X^{A}}}\frac{\partial \rho_{\Omega }\left[X^{A},X^{B}\right]}{\partial t}{\mathrm{dV}}_{X^{B}}=\int_{{\;H}_{X^{A}}}{\;k}_{b}\left[X^{A},X^{B}\right]\left(\frac{\Phi \left[X^{B}\right]-\Phi \left[X^{A}\right]}{{\left|\xi \right|}^{2}}\right){\mathrm{dV}}_{X^{B}}+\int_{{\;H}_{X^{A}}}J_{b}\left[X^{A},X^{B}\right]{\mathrm{dV}}_{X^{B}}$$

Simplifying the left-hand side of Equation (3), we have
(4)$$\int_{{\;H}_{X^{A}}}\frac{\partial \rho_{\Omega }\left[X^{A},X^{B}\right]}{\partial t}{\mathrm{dV}}_{X^{B}}=\frac{\partial \rho_{\Omega }\left[X^{A}\right]}{\partial t}V_{X^{A}}$$
where $V_{X^{A}}$ is the volume of the horizon of point $X^{A}$. Similarly, the charge source term at point $X^{A}$ is obtained by the average of the same in all the bonds attached to $X^{A}$ in its horizon ${\;H}_{X^{A}}$: (5)$$\int_{{\;H}_{X^{A}}}J_{b}\left[X^{A},X^{B}\right]{\mathrm{dV}}_{X^{B}}=J\left[x\right]V_{X^{A}}$$

Therefore, we may write the PD conservation of charge equation by equating Equations (3–5) for any material point $X^{A}$ as: (6)$$\frac{\partial \rho_{\Omega }\left[X^{A}\right]}{\partial t}={\dot{\rho }}_{\Omega }\left[X^{A}\right]=\int_{{\;H}_{x}}k_{e}\left[X^{A},X^{B}\right]\left(\frac{\Phi \left[X^{B}\right]-\Phi \left[X^{A}\right]}{{\left|\xi \right|}^{2}}\right){\mathrm{dV}}_{X^{B}}+J\left[X^{A}\right]$$
where: ${\dot{\rho }}_{\Omega }$ is the time rate of change of total charge, $k_{e}\left[X^{A},X^{B}\right]=\frac{{\;k}_{b}\left[X^{A},X^{B}\right]}{V_{X^{A}}}\mathrm{in}\;\mathrm{case}\;\mathrm{of}\;3-D$ and $\frac{{\;k}_{b}\left[X^{A},X^{B}\right]}{A_{X^{A}}}$ in case of 2D are defined as the microconductivity of the electrical bond  ξ. 

In compact form, Equation (6) is written as follows: (7)$$\frac{\partial \rho_{\Omega }\left[X^{A},t\right]}{\partial t}={\dot{\rho }}_{\Omega }\left[X^{A},t\right]=\int_{H}f_{J}\left(\Phi^{\prime },\Phi ,X^{B},X^{A}\right){\mathrm{dV}}_{X^{B}}+J\left(X^{A},t\right)$$
where $f_{J}\left[X^{A},X^{B}\right]=k_{e}\frac{\varphi \left(X^{A},X^{B}\right)}{{\left|\xi \right|}^{2}},\;\varphi \left(X^{B},X^{A}\right)=\Phi \left[X^{B}\right]-\Phi \left[X^{A}\right],{\;f}_{J}$ is the electric current flow density function, ${\dot{\rho }}_{\Omega }$ the time rate of change density, φ is electric potential difference among $X^{B}{\mathrm{and}\;X}^{A}$, and J is the applied current flux. 

Borrowing the definition of PD heat flux from Florin and Monchai (27) and extending it to obtain the PD electric charge flux equation at any point $X^{A}$, we get
(8)$$j_{\mathrm{PD}}\left[X^{A},t\right]=-\int_{H_{X^{A}}^{+}}k_{e}\left[X^{A},X^{B}\right]\left(\frac{\Phi \left[X^{B}\right]-\Phi \left[X^{A}\right]}{\left|\xi \right|}\right)e\left[X^{A},X^{B}\right]{\mathrm{dA}}_{X^{B}}$$
where $H_{X^{A}}^{+}$ is the particular area in the horizon of point $X^{A}$ with neighboring points of higher electrical potential than that at $X^{A}$.

### 2.2. PD Formulation for Heat Conduction

Since the linearized bond-based model is the special case of the generalized state-based model, we start our bond based formulation based on the following SB–PD equation.
(9)$${\rho C}_{v}\dot{\theta }\left[X^{A},t\right]=\int_{H}\left(\underline{h}\left[X^{A}\right]❬\xi ❭-\underline{h}\left[X^{B}\right]❬-\xi ❭\right){\mathrm{dV}}_{X^{B}}+h_{s}\left(X^{A}\right)$$
where: $\underline{h}\left[X^{A},X^{B}\right]=-\bar{\kappa }\left[X^{A},X^{B}\right]\left(\frac{\theta \left[X^{B}\right]-\theta \left[X^{A}\right]}{\left|\xi \right|}\right)e\left[X^{A},X^{B}\right]$ is heat flow density, $h_{s}$ is the heat source due to volumetric heat generation, θ is temperature, $\bar{\kappa }$ is micro conductivity of thermal bond, and $C_{v}$ is the specific heat capacity.

If the heat flow density associated with two material points $X^{A}$ and $X^{B}$ is a function of the temperature difference between the two points, then the following expression holds true [18,28]:(10)$$\underline{h}\left[X^{A}\right]❬\xi ❭=-\underline{h}\left[X^{B}\right]❬-\xi ❭$$

This leads to the BB–PD equations. In this case, the heat flow density function, $f_{h}\left[X^{A},X^{B}\right]$, is defined as
(11)$$f_{h}\left[X^{A},X^{B}\right]=\underline{h}\left[X^{A}\right]❬\xi ❭-\underline{h}\left[X^{B}\right]❬-\xi ❭=2\underline{h}\left[X^{A}\right]❬\xi ❭$$

So that the PD heat conduction equation can be found as
(12)$${\rho C}_{v}\dot{\theta }\left[X^{A},t\right]=\int_{H}f_{h}\left(\theta^{B},\theta^{A},X^{B},X^{A}\right){\mathrm{dV}}_{X^{B}}+h_{s}\left(X^{A}\right)$$

We established the bond-based PD heat conduction formulation based on the work of Oterkus et al. [28]. Their governing equation takes the same form as Equation (12) with pair-wise heat flow density function $f_{h}$, expressed as
(13)$$f_{h}\left[X^{B},X^{A}\right]=\bar{\kappa }\frac{\tau \left(X^{A},X^{B}\right)}{{\left|\xi \right|}^{2}},\;\tau \left(X^{B},X^{A}\right)=\theta \left[X^{B}\right]-\theta \left[X^{A}\right]$$
where τ is temperature difference among $X^{A}{\;\mathrm{and}\;X}^{B}$.

Moreover, the heat flux vector $\left(q\right)$ in 2sD can be express as:(14)$$q_{\mathrm{PD}}\left[X^{A}\right]=-\int_{H_{x}^{+}}\bar{\kappa }\frac{\theta \left[X^{B}\right]-\theta \left[X^{A}\right]}{\left|\xi \right|}e\left[X^{A},X^{B}\right]{\mathrm{dA}}_{X^{B}}$$

### 2.3. Peridynamic Formulation for the System of Thermoelectric

Overutilization of fossil fuel, the world’s energy demand, and climate change are the main factors that push forward the research in the area of new energy materials. The main criteria of these materials are mainly sustainability and environmental friendliness. In this context, thermoelectric materials (TEMs) are excellent candidates due to their simplicity, compactness, scalability, sustainability, portability, reliability, and environmental friendliness. Moreover, TEMs are quite capable of converting temperature change to electric voltage or electricity to heat. The first scenario couples electric field and temperature gradient and is known as the Seebeck effect, while the second scenario is known as the Peltier effect which is the reverse of the first one. The Seebeck effect was discovered by Seebeck in 1821 and the Peltier effect by Peltier in 1855, and these two effects are commonly called the thermoelectric effect [44]. Therefore, we dedicate this section to the development of peridynamic governing equations for thermoelectricity. These equations are described as constitutive equations and the conservation laws of thermoelectricity. The detailed description and derivations of these equations using classical approach may be found in [45]. 

#### 2.3.1. Peridynamic Constitutive Equations

Here, we write the PD constitutive equation due to the presence of both Seebeck and Peltier effects at the same time. The presence of the Seebeck effect is due to temperature gradient, and that of Peltier effects is due to the change in an electric current. Thus, the PD constitutive equation for the system of coupled thermoelectric is obtained by extending the approach developed by Agwai [18] as follows.
(15)$$q\left(X^{A},t\right)=-{\alpha \;\theta \;k}_{b}\frac{\Phi \left(X^{B},t\right)-\Phi \left(X^{A},t\right)}{||\xi ||}-\left(\hat{\kappa }+\alpha^{2}{\theta \;k}_{b}\right)\frac{\Theta \left(X^{B},t\right)-\Theta \left(X^{A},t\right)}{||\xi ||}$$
where $q,\;\alpha ,\theta ,\;\Phi ,k_{b}$, and $\hat{\kappa },$ respectively, are heat flux, Seebeck coefficient, temperature, electric potential, PD conductivity of electrical bonds, and thermal bonds conductivity.

Decoupling Equation (15) results in the PD thermal conduction equation as proposed by [18,26,27]. The PD constitutive equation for the electric current is obtained by combining Ohm’s law and Seebeck effect. The presence of thermal and electric fields simultaneously results in the generalized thermoelectric effect. Moreover, the PD constitutive equation for the electric current, by taking in to account the Seebeck and Ohm effects, is given by: (16)$$j\left(X^{A},t\right)=-{\;k}_{b}\frac{\Phi \left(X^{B},t\right)-\Phi \left(X^{A},t\right)}{||\xi ||}e\left[X^{A},X^{B}\right]-{\;k}_{b}\alpha \;\frac{\Theta \left(X^{B},t\right)-\Theta \left(X^{A},t\right)}{||\xi ||}e\left[X^{A},X^{B}\right]\;\;$$
where j is the PD current flow and $||\xi ||$ is the magnitude of the bond vector. Uncoupling Equation (16) results in the PD equation of electric conduction as proposed by Prakash and Seidel (37).

#### 2.3.2. Peridynamic Balance Laws

The PD energy balance equation that governs the system of thermoelectric is described as,
(17)$$\rho_{0}C_{v}\dot{\Theta }\left(X^{A},t\right)=\int_{H_{X^{A}}}\left(\;\alpha \theta k_{e}\frac{\Phi \left(X^{B},t\right)-\Phi \left(X^{A},t\right)}{||\xi ||}+\left(\bar{\kappa }+\alpha^{2}\theta k_{e}\right)\frac{\Theta \left(X^{B},t\right)-\Theta \left(X^{A},t\right)}{||\xi ||}\;\right){\mathrm{dV}}^{B}$$
$\mathrm{where}\;\bar{\kappa }=\;\hat{\kappa }/V_{H_{X^{A}}}{\mathrm{and}\;k}_{e}={\;k}_{b}/V_{H_{X^{A}}}$, respectively, are micro-conductivity of thermal bonds and electrical bonds. $V_{H_{X^{A}}}$ is volume of the horizon of nodal point at $X^{A}$. Hence, Equation (17) represents the law of conservation of energy for system of thermoelectric without the heat source term. Considering the source term, the PD thermal energy balance equation for point $X^{A}$ is obtained as: (18)$$\rho_{0}C_{v}\dot{\Theta }\left(X^{A},t\right)=\int_{H_{X^{A}}}\left({\alpha \theta k}_{e}\;\frac{\Phi \left(X^{B},t\right)-\Phi \left(X^{A},t\right)}{||\xi ||}+\left(\bar{\kappa }+\alpha^{2}\theta k_{e}\right)\frac{\Theta \left(X^{B},t\right)-\Theta \left(X^{A},t\right)}{||\xi ||}\;\right){\mathrm{dV}}^{B}+\rho_{0}(X^{A})r\left(X^{A},t\right)$$
where: Cv is the specific heat.

Similarly, the PD balance of electric charge for system of thermoelectric is expressed as: (19)$${\dot{\rho }}_{\Omega }\;\left(X^{A},t\right)=\int_{H_{X^{A}}}\left(k_{e}\;\frac{\Phi \left(X^{B},t\right)-\Phi \left(X^{A},t\right)}{||\xi ||}+k_{e}\alpha \frac{\Theta \left(X^{B},t\right)-\Theta \left(X^{A},t\right)}{||\xi ||}\;\right){\mathrm{dV}}^{B}\;+\rho_{0}\left(X^{A}\right)R\left(X^{A},t\right)$$

The PD properties may be related with the classical material properties by borrowing the expressions from [28] and extend it to electric field. Thus, the electrical and thermal micro-conductivity expressions for one- and two-dimensional analysis are given, respectively, as:(20)$$k_{e}=\frac{2K_{e}}{{A\delta }^{2}},\bar{\kappa }=\frac{2\kappa }{{A\delta }^{2}}\;\mathrm{for}\;\left(1-D\right)\;k_{e}=\frac{6K_{e}}{{\pi h\delta }^{3}},\bar{\kappa }=\frac{6\kappa }{{\pi h\delta }^{3}}\;\mathrm{for}\;\left(2-D\right)$$
where $\delta ,\;h,\kappa ,{\;K}_{e},\;\mathrm{and}\;A$, respectively, denote horizon radius, thickness, thermal conductivity, electrical conductivity, and cross-sectional area. Table 1 below shows the summary of comparison between PD and classical thermoelectric equations.

## 3. Numerical Procedures

PD-based heat conduction for the system of thermoelectric may be solved by utilizing numerical techniques. Here, the domain is subdivided into equally spaced divisions for the sake of simplicity. Finite sum has been used to replace nonlocal integral equation Equation (18) as follows:(21)$$\rho_{\left(A\right)}C_{v}{}_{\left(A\right)}{\dot{\Theta }}_{\left(A\right)}^{n}=\underset{B\in H_{X^{A}}}{\sum }\left({\alpha \;\theta k}_{e}\;\frac{\Phi^{n}\left(X^{B},t\right)-\Phi^{n}\left(X^{A},t\right)}{||\xi ||}+\left(\bar{\kappa }+\alpha^{2}\theta k_{e}\right)\frac{\Theta^{n}\left(X^{B},t\right)-\Theta^{n}\left(X^{A},t\right)}{||\xi ||}\right)V^{B}$$

In Equation (21), n denotes the number time steps, A denotes our point of interest, and B denotes the points within the horizon of A.

$V^{B}$ denotes volume associated with node B. For PD model, the forward difference numerical scheme is applied. After employing the forward difference scheme, the subsequent equation is answered.
(22)$$\Theta_{\left(A\right)}^{n+1}=\frac{∆t}{\rho_{\left(A\right)}C_{v}{}_{\left(A\right)}}\left[\sum_{B\in H_{X^{A}}}\left({\alpha \theta \;k}_{e}\;\frac{\Phi^{n}\left(X^{B},t\right)-\Phi^{n}\left(X^{A},t\right)}{||\xi ||}+\left(\bar{\kappa }+\alpha^{2}\theta k_{e}\right)\frac{\Theta^{n}\left(X^{B},t\right)-\Theta^{n}\left(X^{A},t\right)}{||\xi ||}\right)V^{B}\right]$$

### Modeling of Insulated Crack

Estimation of crack growth is one of the common problems in engineering. The existence of cracks in material inhibits the transfer of heat partially or completely. In peridynamics, heat conduction is interrupted if the bonds are broken as shown in Figure 2, and hence there will be no interaction between the material points. Therefore, the PD thermal conduction for the system of thermoelectric may be revised by introducing $\mu \left(\xi ,t\right)$, the scalar value function, for every bond as follows [27]: (23)$$\rho_{0}C_{v}\dot{\Theta }\left(X^{A},t\right)=\int_{H_{X^{A}}}\mu \left(\xi ,t\right)\left(\bar{\kappa }\frac{\Theta \left(X^{B},t\right)-\Theta \left(X^{A},t\right)}{||\xi ||}\;\right){\mathrm{dV}}^{B}$$
where: (24)$$\mu \left(\xi ,t\right)=\left\{\begin{array}{c} 1\;\;\mathrm{if}\;\mathrm{there}\;\mathrm{is}\;\mathrm{an}\;\mathrm{interaction}\;\mathrm{among}\;X^{A}\;\mathrm{and}\;X^{B}\\ 0\;\;\mathrm{if}\;\mathrm{there}\;\mathrm{is}\;\mathrm{no}\;\mathrm{interaction}\;\mathrm{among}\;X^{A}\;\mathrm{and}\;X^{B}\; \end{array}\right.$$

In this paper, the boundary conditions considered are the Dirichlet (temperature) or Neumann (heat flux) boundary conditions, even though it is also possible to impose convection and radiation conditions. Hence the boundary temperature $T_{b}$ is imposed at the boundary region $\;R_{t}$, by specifying the temperature of the material points on the same region as follows:(25)$$T_{i}=T_{b},\;\;\;\;i\in R_{t}$$

Similarly, the boundary heat flux $q_{b}$ is imposed at the boundary region $\;R_{t}$ by specifying the heat flux of the material points on this region as follows:(26)$$q_{i}=q_{b},\;\;\;\;i\in R_{t}$$

If the boundaries are well insulated, the heat flux through the boundary is assumed to be zero, and hence, the Neumann boundary condition at the boundary region $R_{t}$ is given as follows:(27)$$q_{i}^{N}=0,\;\;\;\;i\in R_{t\;}$$
wherew N depicts the normal direction of boundary surface, and we call such type of boundary a free boundary. 

## 4. Results and Discussions 

### 4.1. Validation of PD Theory 

In this article, we applied the bond-based PD approach to show the practicality of the proposed formulation. Three examples are presented; the first one illustrates two-dimensional heat conduction phenomena by taking into account non-symmetric and symmetric boundaries. The second one demonstrates the linear uncoupled Seebeck effect for thermoelectric phenomena; here, we considered both constant material properties and temperature-dependent material properties. Finally, the ability of PD theory in the treatment of insulated cracks has been presented. 

Example 1: Two-dimensional heat conduction (uncoupled case, α = 0) 

To prove the capability of the PD scheme, we present the decoupled two-dimensional heat conduction. The geometrical and material parameters are specified in Table 2 below. Here, we considered two different boundary conditions, non-symmetric and symmetric boundaries. In the case of non-symmetric boundary, the following initial and boundary conditions were considered. 

Case 1: Non-symmetric Boundary

Initial Conditions: (28)$$\Theta \left(x,y,0\right)=0\;{}^{\circ}C,-\frac{L}{2}\leq x\leq \frac{L}{2},-\frac{W}{2}\leq y\leq \frac{W}{2},$$

Boundary Conditions: (29)$$\Theta \left(\frac{W}{2},y,t\right)=100\;{}^{\circ}C\;\left(\mathrm{Top}\right),\;\Theta \left(-\frac{W}{2},y,t\right)=0\;{}^{\circ}C,\left(\mathrm{Bottom}\right)$$

Here, the domain is divided into 20 nodal points in the x and 20 in the y direction. The spacing among the points is 0.1 cm and the time step is $10^{-4}$ s. Figure 3a illustrates analytical and PD solutions of temperature variations in two dimensions. Results obtained from the above two solutions witnessed their close agreements. Figure 3b shows the temperature contour in the case of non-symmetric boundary condition for time t = 20 s, 40 s, 100 s, and 200 s.

Case 2: Symmetric Boundary 

In this case, we considered the following boundary conditions.
(30)$$\Theta \left(\frac{W}{2},y,t\right)=100\;{}^{\circ}C,\;\Theta \left(-\frac{W}{2},y,t\right)=100\;{}^{\circ}C,$$

Initially, we considered the temperature in the plate was 0 °C, and once 100 °C is imposed on the top and bottom boundaries, we observed the rise in temperature inside the plate. As shown in Figure 4a, the temperature increases with time and attains its steady-state value. Figure 4b shows the temperature contour in the case of symmetric boundary condition for time t = 20 s, 40 s, 100 s and, 200 s. 

Example 2: Heat conduction of two-dimensional plate (linear uncoupled Seebeck effect)

This example presented the comparison between our peridynamics solution and analytical solution for coupled thermoelectric phenomena. Here, we considered a single pellet of bismuth telluride $\left({\mathrm{Bi}}_{2}{\mathrm{Te}}_{3}\right).$

Case 1: Considering constant material properties

Geometrical parameters and material properties and are shown in Table 3 below [47]. To demonstrate the usefulness of the PD method, we further considered the thermoelectric element shown in Figure 5 below. In this case, we technically forced the problem as a one-dimensional and linear problem. The electric flux is zero on the boundaries due to isolation; hence there are no Thompson and Peltier effects. The main goal of this sample example is to determine the temperature and electric potential in addition to the heat flux. 

In this example, the domain is divided into 76 particles in the x direction and 69 in the y direction. Boundary conditions can be imposed as temperature and voltage. The thermo-element shown in Figure 5 is grounded at its right lower corner where voltage is specified as zero [46]. The temperatures on the left and right boundaries are imposed as $T_{b}$ on the region $\;R_{t}$, and the temperature of particles in this region should be specified as follows:(31)$$T_{i}=T_{b},\;\;\;\;i\in R_{t}$$

The top and bottom boundaries $R_{t}$ are well insulated; hence the heat flow through the boundary is assumed to be zero, therefore:(32)$$q_{i}^{N}=0,\;\;\;\;i\in R_{t}$$
where N is the normal direction to the boundary surface. 

Since thermal and electric fields are scalar by nature, temperature and voltage need to be prescribed at least at one point [47].

Boundary Condition:(33)$$\Theta \left(0,t\right)=273{}^{\circ}K,\;\Theta \left(L,t\right)=298{}^{\circ}K,\;V\left(L\right)=0\;v,$$

Figure 6a below shows the temperature variation comparison between our peridynamic results with that of an analytical solution for the case of constant material properties. As can be seen from Figure 6a, our PD result is in very good agreement with that of the analytical result. Furthermore, Figure 6b shows the temperature distribution of bismuth telluride $\left({\mathrm{Bi}}_{2}{\mathrm{Te}}_{3}\right)$ pellet. 

The temperature and voltage distributions may be obtained analytically as follows [46]:(34)$$T\left(x\right)=273+16404x,$$
where T is in Kelvin.
(35)$$V\left(x\right)=3.033\left(L-x\right),$$
(36)$$q=-\kappa \left(T_{a}\right)\frac{∆T}{l},$$
(37)$$T_{a}=\frac{T_{h}+T_{c}}{2}=285.5{}^{\circ}K\;$$
(38)$$∆T=T_{h}-T_{c}=25{}^{\circ}K,$$

Figure 7 below shows the comparison between the exact values of the heat flux with that of the peridynamic result in the case of constant material property. As can be observed from Figure 7, results from PD and steady-state solution are in close agreement after 2 s, and the flux distribution along the X coordinate is constant.

The analytical procedure for the thermal flux is obtained as follows [47]:(39)$$q=-\kappa \left(T_{a}\right)\frac{∆T}{l}=27910W/m^{2},$$

Similarly, Figure 8 shows the electric potential variation comparison between the PD results and analytical solution in the case of constant material properties. Figure 8a witnessed that our PD result is in very good agreement with that of the analytical solution. Figure 8b shows the electric potential distribution of bismuth telluride $\left({\mathrm{Bi}}_{2}{\mathrm{Te}}_{3}\right)$ pellet. 

We furthermore compare the peridynamic results with results from the literature. Table 4 below summarizes the comparison between our PD results and the results from the references [46,47]. It is observed that our result is in very close agreement with results from the references. Hence, our peridynamic model captures temperature and electric potential distribution well.

Case2: Considering temperature dependence of material properties 

In this example, we present the comparison between the peridynamic solution and analytical solution for coupled thermoelectric phenomena. Table 5 shows the geometrical dimensions and material properties that are temperature-dependent [47]. In this example, we found the temperature and electric potential distribution. Boundary conditions and other assumptions are similar to the previous example. 

Figure 9 shows the temperature variation comparison between the peridynamic results with the analytical solution for the case of temperature-dependent material properties. As can be seen from Figure 9, our PD result is in very good agreement with the analytical result. 

The analytical solution for temperature and voltage distributions in the case of temperature-dependent material properties may be obtained as follows [47]:(40)$$V\left(x\right)=\sqrt{0.0181-1.41x}+2.36x-0.13\;,$$
(41)$$T\left(x\right)=684.8-\sqrt{169587-1.31\times 10^{7}x}\;,$$
(42)$$q=\frac{C_{1}}{2a}\;,$$
(43)$$C_{1}=\frac{\kappa {\left(T_{h}\right)}^{2}-\kappa {\left(T_{c}\right)}^{2}}{l},\;a=-4.26\times 10^{-3}\;,$$

Figure 10 below shows the electric potential variation comparison between our peridynamic results and analytical solution for the case of temperature-dependent material properties. Figure 10 witnessed that our PD result closely agreed with that of the analytical solution.

Figure 11 below shows the comparison between the exact values of the heat flux with that of our peridynamic result in the case of temperature-dependent material property. As can be observed from Figure 11, results from PD and steady-state solution are in good agreement, and the flux distribution along the x coordinate is constant.

In addition to comparing our PD result with the analytical solution, we compared our results with the results from the references [46,47]. Table 6 below summarizes this comparison.

### 4.2. Examples on Thermoelectric Plate with Stationary and Moving Crack

In this section, the following examples will be considered to show the applicability of the peridynamic formulation in capturing discontinuities. 

Example 3: Edge Crack Domain under constant flux

In this example, a square plate of 2 cm ×2 cm made from TEMs with an edge-insulated crack of length of  2a=1.0 cm has been analyzed. In this example, the following two different boundary conditions are considered. 

Case 1: Top and bottom boundaries are insulated 

Here, the left edge is subjected to a constant temperature of −100 °C, and in the meantime, the temperature of the right side edge is maintained at  100 °C. The top and bottom edges are assumed to be well insulated. The geometry and the boundary conditions are shown in Figure 12 below. As a result of the above boundary conditions, the domain is exposed to a constant heat flux. Geometric dimensions and material properties are presented in Table 7. In this example, the domain is segmented into 101 material points in the x and 101 material points in the y direction. 

To demonstrate the capability of the PD model for thermal loading, a comparison between the peridynamic solution and the solution from general FEM code ABAQUS has been made. As can be seen from Figure 13, the PD result and the result from FEM code ABAQUS are in good agreement. Furthermore, it is observed that the temperature field is not affected due to the presence of the crack.

Case 2: Left and right boundaries are insulated

The same square plate of 2 cm ×2 cm with an edge crack of length 2a=1.0 cm is subjected to a temperature of −100 °C on its bottom edge and  100 °C on its top edge. The left and right edges are assumed to be well insulated. The conditions are shown in Figure 14 below. The discretization and material properties are the same as in the previous example.

To demonstrate the validity of the PD model, finite element solutions are also presented for verification purpose. The results from PD and from FEM code ABAQUS are shown in Figure 15 below, and they are found to be in good agreement. Furthermore, it is observed from Figure 15 that the temperature field is considerably affected due to the presence of crack, unlike the previous example. Figure 16 also shows close results along x = −0.5. From this result, we may conclude that our peridynamic formulation is in good agreement with that of the finite element result. 

Example 4: Heat conduction with a moving crack 

To substantiate the novelty of our PD model, we provide here an example that demonstrates the beauty of PD in treating propagating cracks. The dimensions, number of material points, and material properties are considered to be the same as the one in Example 3, except the crack in this example is a center crack of length *2a*. Here, we assume an initial crack with a size of $a_{o}=1.25\;\mathrm{mm}$ starting from point A (x = −0.5 cm) as shown in Figure 17 below. The bottom and top edges of the plate are exposed to a temperature of −100 °C and 100 °C, respectively, while the right and left edges are kept insulated. 

The crack initiates with a length of 1.25 mm and is expected to travel along the X direction as in Figure 17. Here, the crack travels at a velocity of 0.0125 cm/s as it approaches point B(x = 0.5 cm), and hence ensuing length of the crack is measured to be 1 cm. 

Figure 18 below shows our PD results in the case of moving cracks. In this particular example, it is not possible to compare our PD results with that of FEM due to the fact that FEM requires new models for each and every step of simulation, and hence it is time-consuming and cumbersome. Thus, it is inspiring to conclude at this point that peridynamic formulation is one of the promising computational tools in handling growing crack.

## 5. Conclusions

Electric charge and heat transport in the realm of PD for the system of thermoelectric has been presented in this paper. The electrical potentials and temperature gradients of the local theory have been replaced with the functional integral of electrical potential fields and temperature. The functional integrals are effective whether we have evolving discontinuities or not. 

To establish the ability of PD, numerous sample problems have been solved and compared with results from finite element solutions, from analytical solutions or from literature. The first example demonstrates 2D heat conduction for the decoupled thermoelectric plate. In this sample problem, both symmetric and non-symmetric boundary conditions were considered, and comparisons have been made with that of the analytical solutions. We observed that results from the analytical and PD were closely agreed. In the second sample problem, heat diffusion of 2D thermoelectric plate with linear uncoupled Seebeck effect has been considered. In this example, two cases have been analyzed; constant material properties and temperature-dependent material properties. In this example results of analytical solution have been compared with PD. Here the results indicated that our PD results are closely agreed with analytical solutions. Moreover, we compared our PD results of temperature and electric potentials with those of the literature [46,47]. As we can see from Table 4 and Table 6, the results are in good agreement. 

This paper also verified the capability of PD in the treatment of stationary and moving discontinuities. The third and fourth examples demonstrated the capability of PD in capturing discontinuities. By comparing the PD results with those from FEM, the accuracy of PD theory in capturing transient heat conduction phenomena on thermoelectric plate with insulated stationary crack has been demonstrated. It was witnessed that good agreements were obtained between results from PD and finite element solutions. Therefore, we can conclude that PD theory is capable of handling discontinuities. Furthermore, the formulation developed in this study will be used in the future to capture thermal shock response of materials with insulated or permeable cracks of arbitrary shapes.

## Figures and Tables

**Figure 1 materials-13-02546-f001:**
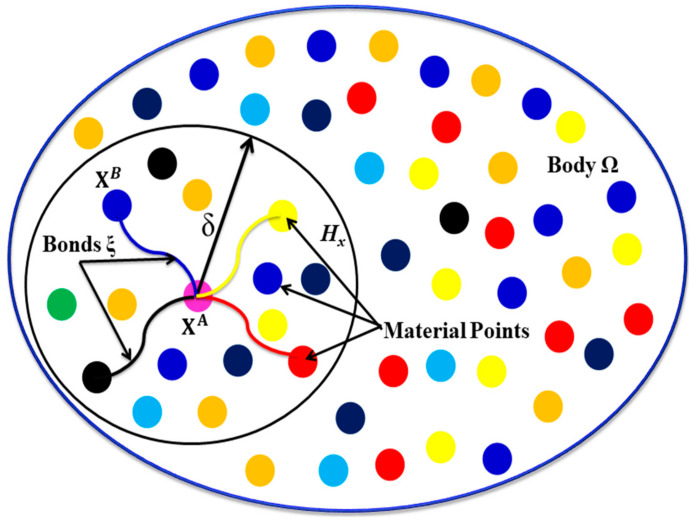
Peridynamic representation of problem domain; here, point $X^{A}$ is influenced by all points within its horizon. $H_{X^{A}}$ is the horizon of $X^{A}$, and δ is the radius of the horizon.

**Figure 2 materials-13-02546-f002:**
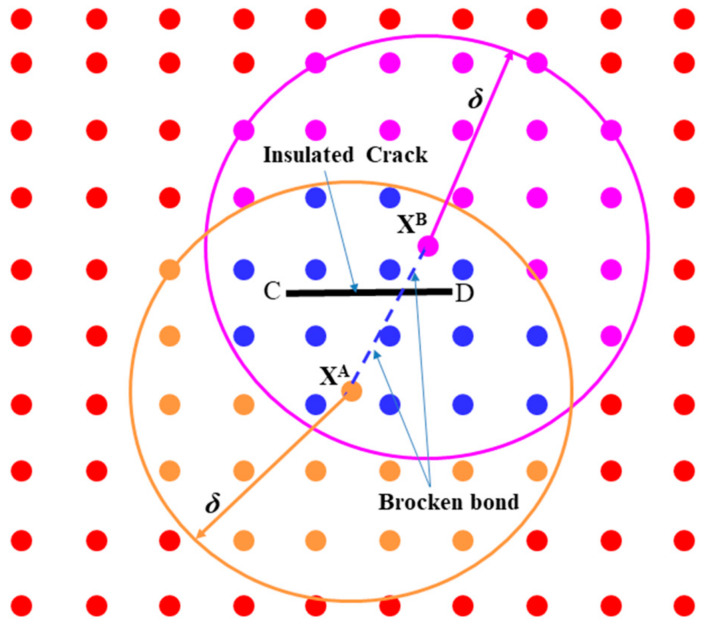
Breakage of bond between nodal points $X^{A}$ and $X^{B}$ that intersect an insulated crack [24]

**Figure 3 materials-13-02546-f003:**
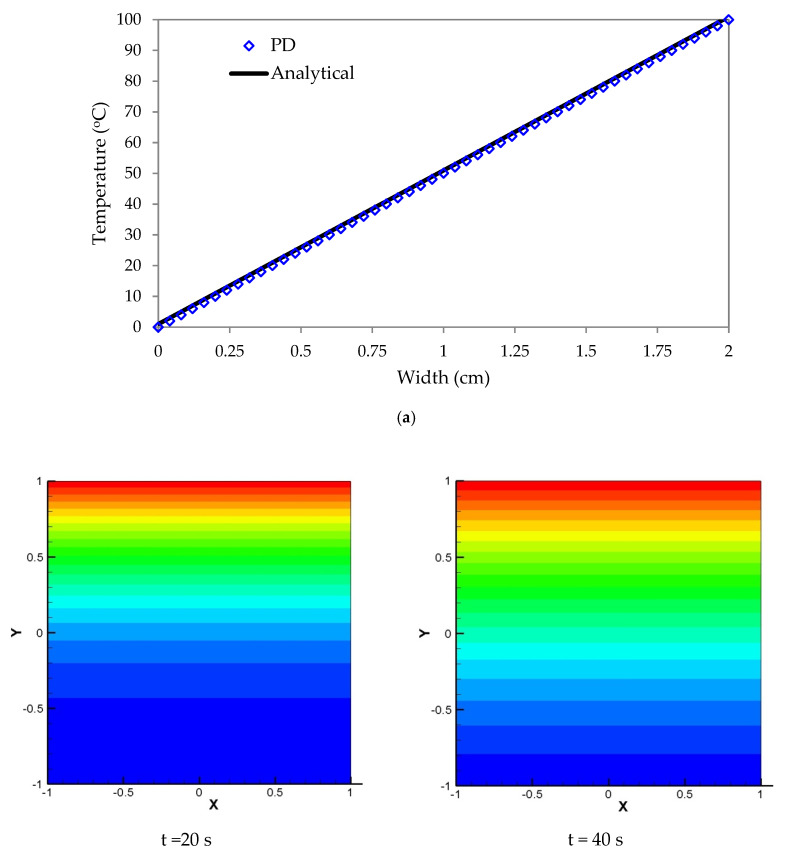
(**a**) 2D temperature variations from peridynamic (PD) and analytical solutions, (**b**) Temperature distribution of 2D plate for non-symmetric boundary (PD).

**Figure 4 materials-13-02546-f004:**
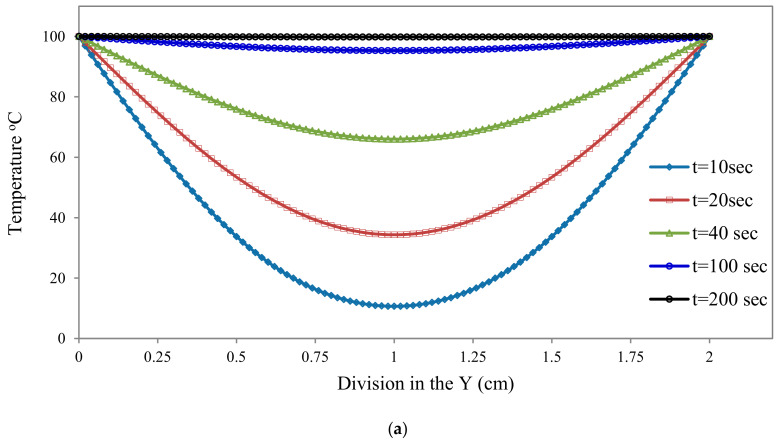
(**a**) 2D temperature variations from PD for symmetric boundary, (**b**) Temperature distribution of 2D plate for symmetric boundary condition.

**Figure 5 materials-13-02546-f005:**
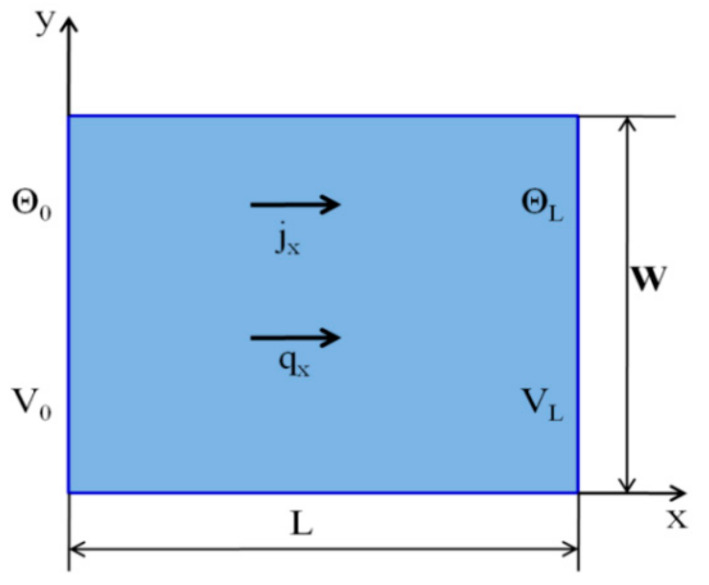
Geometric Parameters and boundary condition for thermoelectric element.

**Figure 6 materials-13-02546-f006:**
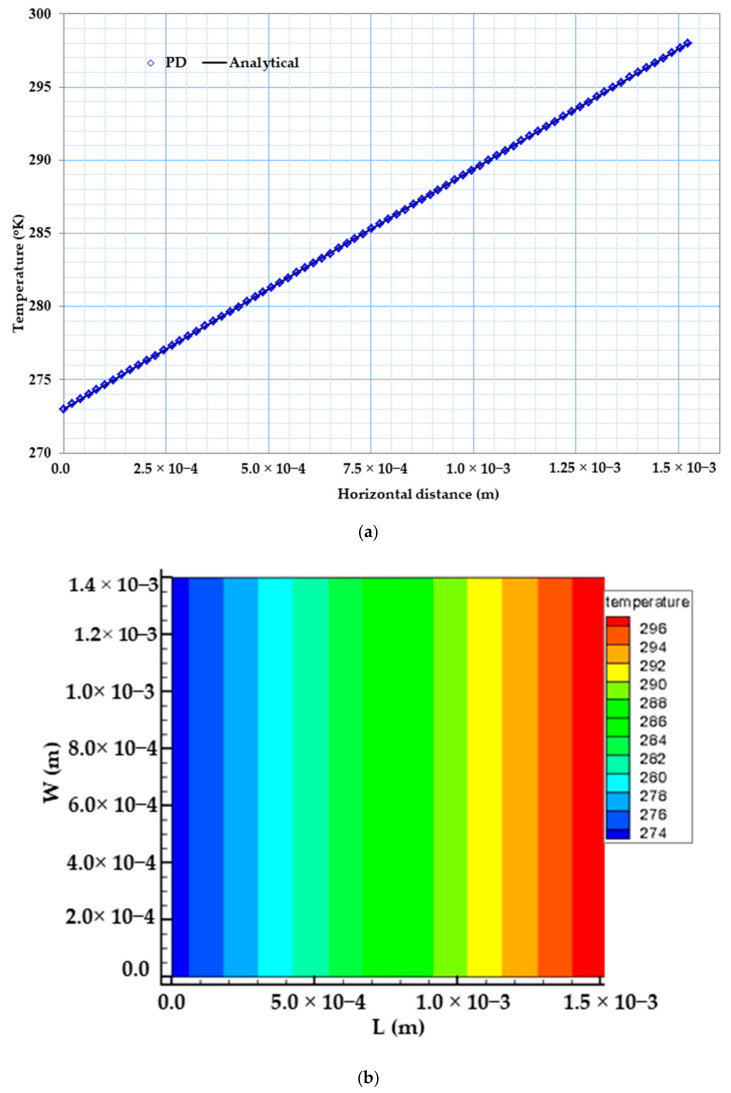
(**a**) 2D temperature variations from PD and analytical solutions for coupled case, (**b**) Temperature (K) distribution of bismuth telluride pellet.

**Figure 7 materials-13-02546-f007:**
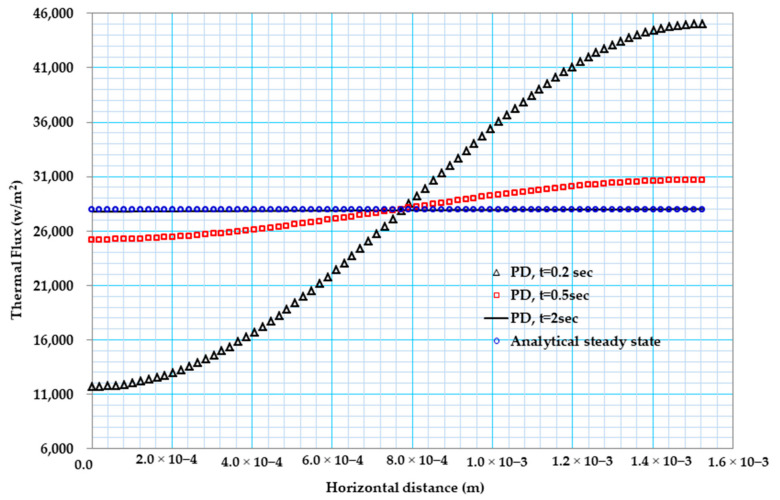
Heat flux (w/m2) distribution of bismuth telluride pellet.

**Figure 8 materials-13-02546-f008:**
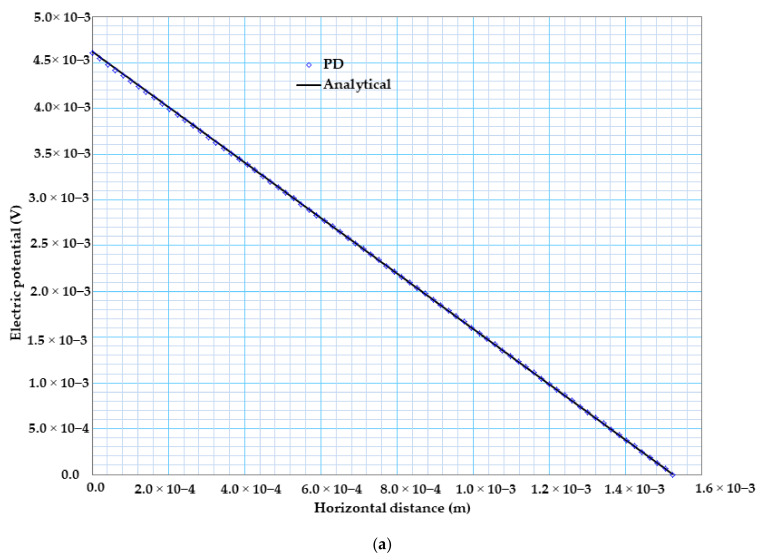
(**a**) 2D electric potential variations from PD and analytical solutions for constant material properties. (**b**) Electric potential distribution of bismuth telluride pellet.

**Figure 9 materials-13-02546-f009:**
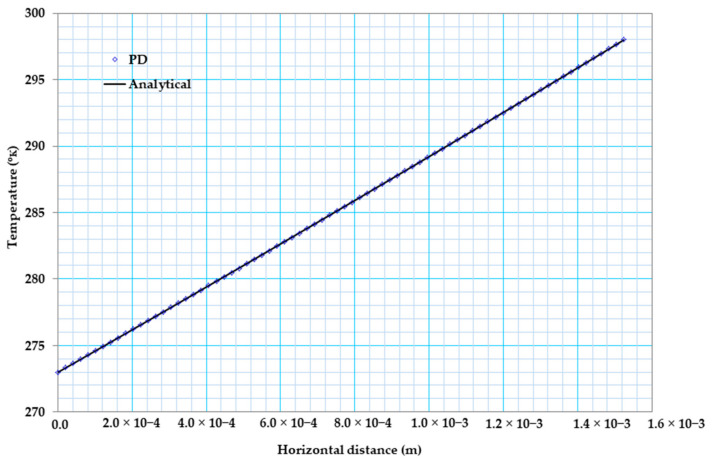
2D temperature variations from PD and analytical solutions for temperature-dependent material properties.

**Figure 10 materials-13-02546-f010:**
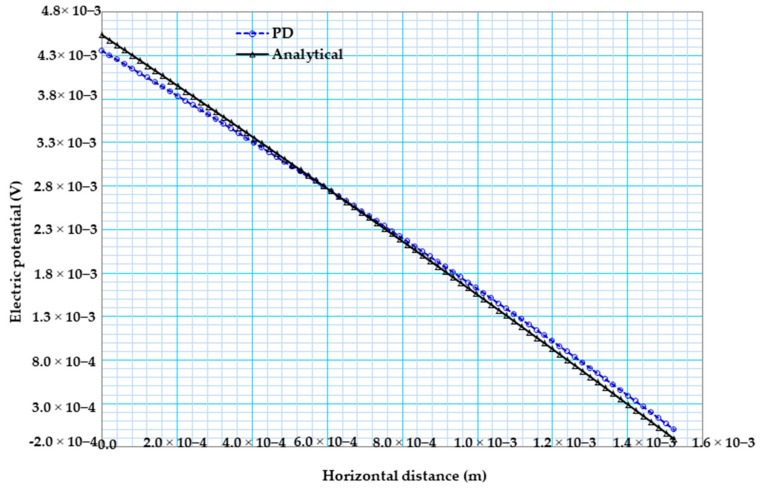
2D electric potential variations from PD and analytical solutions for temperature-dependent material properties.

**Figure 11 materials-13-02546-f011:**
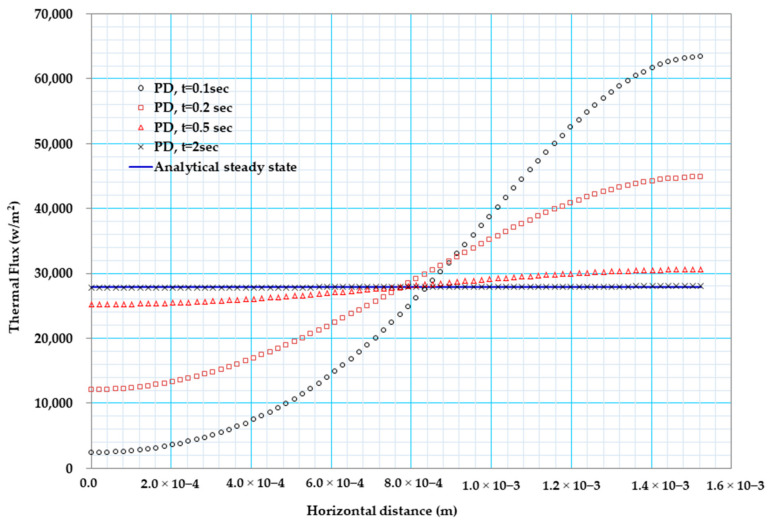
Heat flux (w/m2) distribution of bismuth telluride pellet.

**Figure 12 materials-13-02546-f012:**
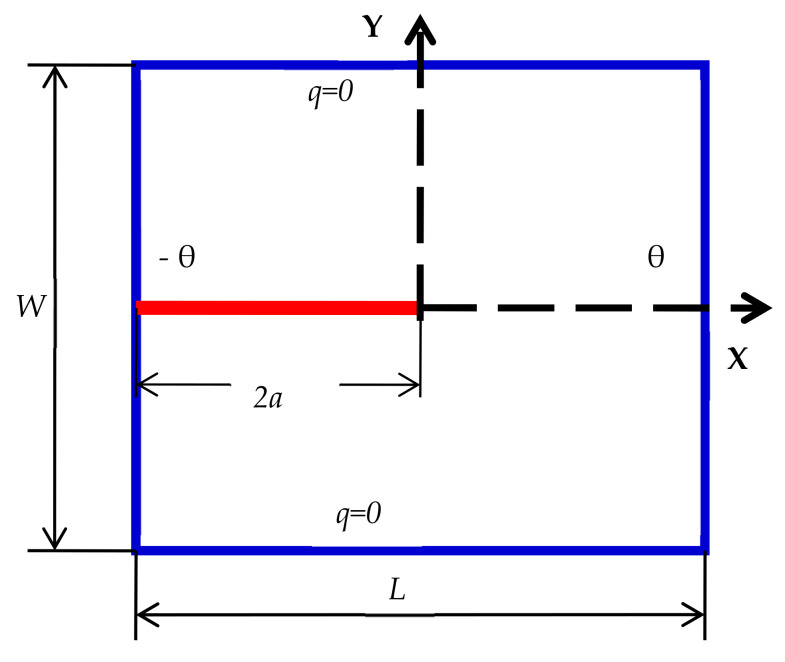
Edge crack (a) geometric parameters and boundary condition.

**Figure 13 materials-13-02546-f013:**
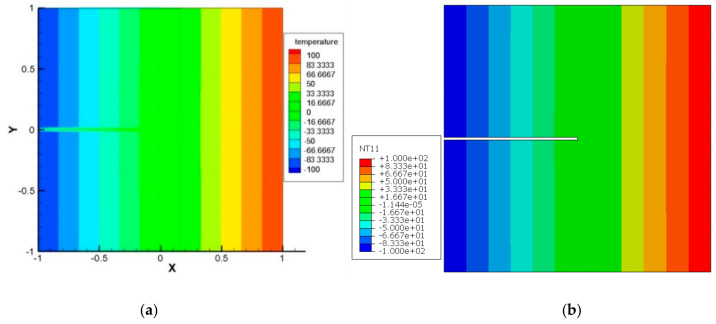
Temperature contour in case of edge crack, at t = 100 s. (**a**) PD result; (**b**) FEM result

**Figure 14 materials-13-02546-f014:**
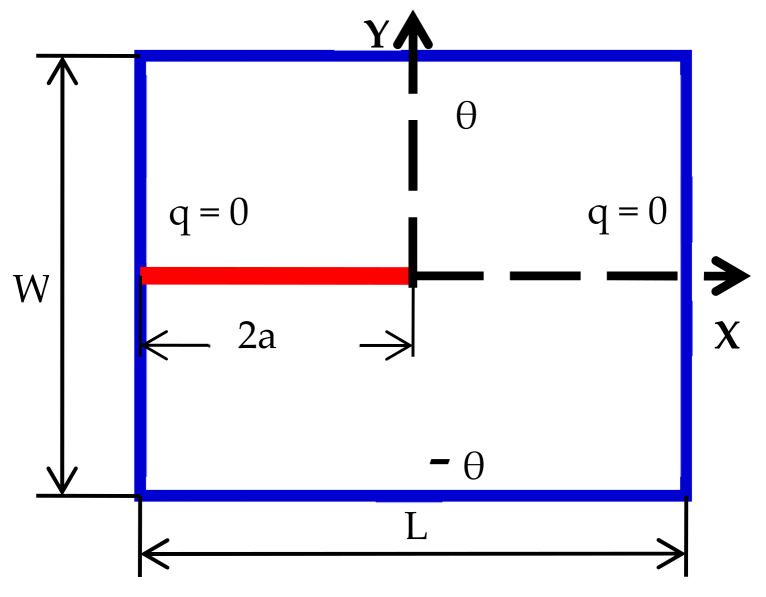
Edge crack (b) geometry and the boundary conditions of a square plate with an edge crack.

**Figure 15 materials-13-02546-f015:**
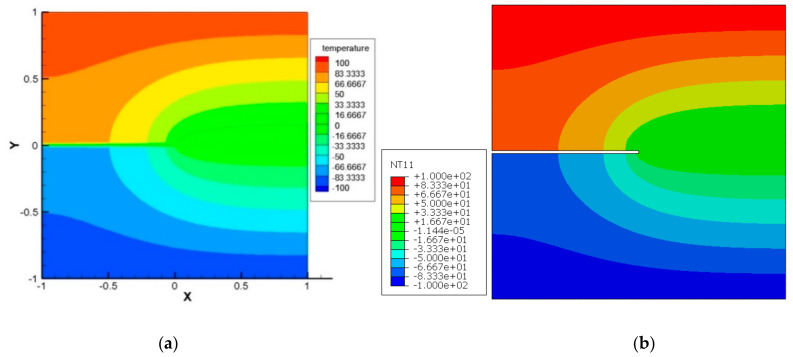
Temperature contour in case of edge crack at t = 100 s. (**a**) PD result, (**b**) FEM result

**Figure 16 materials-13-02546-f016:**
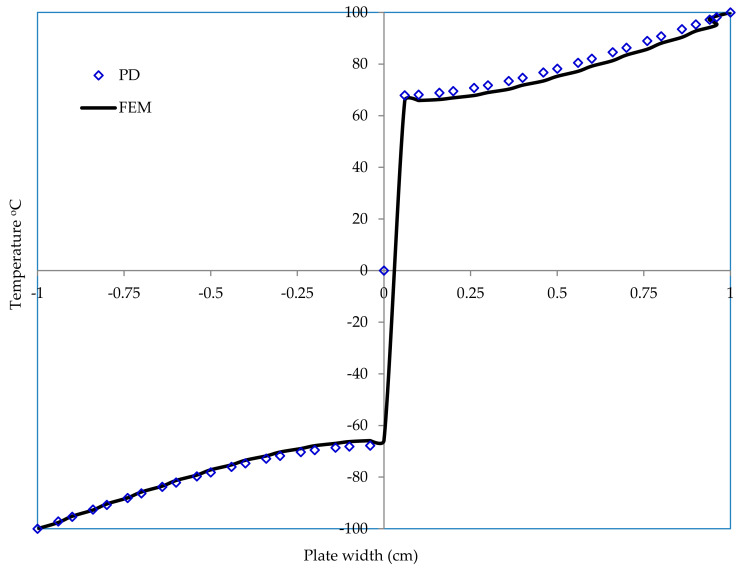
Temperature distribution along x = −0.5.

**Figure 17 materials-13-02546-f017:**
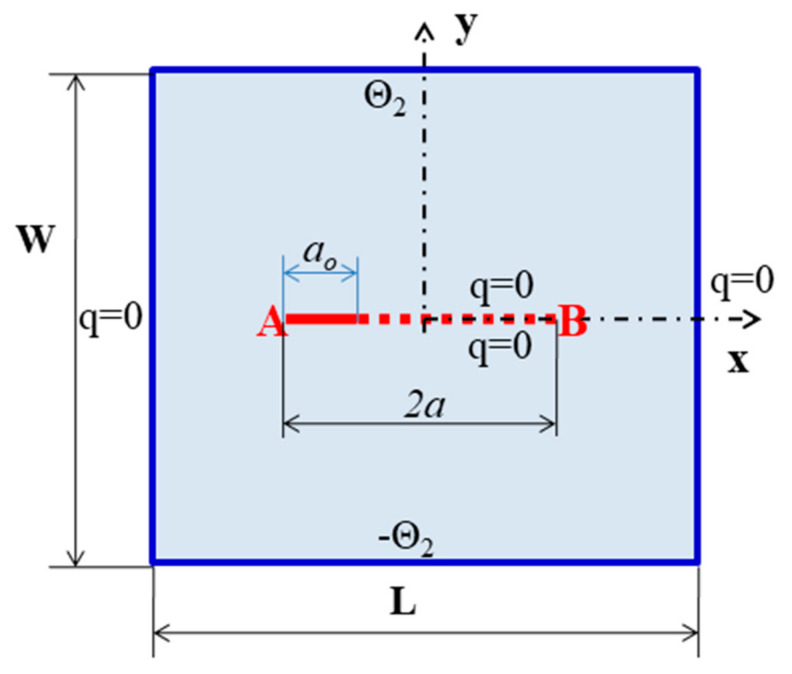
Geometry and the boundary conditions of Example 4.

**Figure 18 materials-13-02546-f018:**
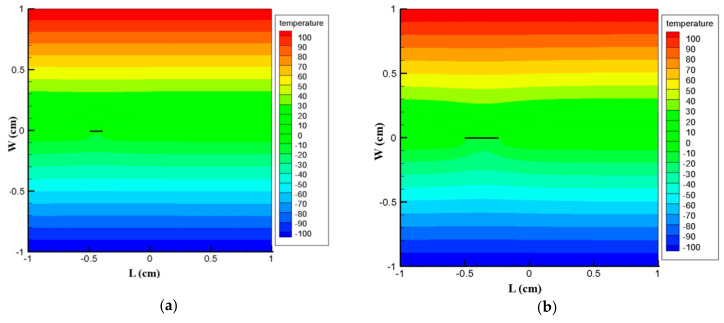
Temperature contour of plate with crack. (**a**) Temperature contour (°C) at t = 10 s; (**b**) Temperature contour (°C) at t = 20 s; (**c**) Temperature contour (°C) at t = 40 s; (**d**)Temperature contour (°C) at t = 50 s; (**e**) Temperature contour (°C) at t = 60 s; (**f**) Temperature contour (°C) at t = 80 s.

**Table 1 materials-13-02546-t001:** Summary of comparison between PD and Classical Equations (Local Vs. PD formulation for TEM).

Physical Problem	Heat Conduction	Electrical Conduction
Conservation principles	Conservation of energy	Conservation of electric charge
Classical Equations	${\rho C}_{v}\dot{\theta }=\kappa \nabla^{2}\theta +j^{2}\gamma_{E}-j\cdot \left(\nabla \beta -\alpha \nabla \theta \right)$	$\rho \dot{z}=\nabla_{x}\cdot \left(\kappa_{E}\nabla \Phi \right)+\nabla_{x}\cdot \left(\kappa_{E}\alpha \nabla \theta \right)+\rho R\left[x\right]$
Peridynamic Equations	$\rho_{0}C_{v}\dot{\Theta }\left(X^{A},t\right)=\int_{H_{X^{A}}}\left(\;\alpha \theta k_{e}\frac{\Phi \left(X^{B},t\right)-\Phi \left(X^{A},t\right)}{||\xi ||}+\left(\kappa ¯+\alpha^{2}\theta k_{e}\right)\frac{\Theta \left(X^{B},t\right)-\Theta \left(X^{A},t\right)}{||\xi ||}\;\right){\mathrm{dV}}^{B}$	${\dot{\rho }}_{\Omega }\;\left(X^{A},t\right)=\int_{H_{X^{A}}}\left(k_{e}\;\frac{\Phi \left(X^{B},t\right)-\Phi \left(X^{A},t\right)}{||\xi ||}+k_{e}\alpha \frac{\Theta \left(X^{B},t\right)-\Theta \left(X^{A},t\right)}{||\xi ||}\;\right){\mathrm{dV}}^{B}+\rho_{0}\left(X^{A}\right)R\left(X^{A},t\right)$
Constitutive Equations	Generalized Fourier’s law	Generalized Ohm’s law
Classical Equatons	Generalized Fourier’s law q=−κ∇θ+αθ j	Generalized Ohm’s law $j=-\kappa_{E}\nabla \Phi -\kappa_{E}\alpha \nabla \theta$
Peridynamic Equations	$q\left(X^{A},t\right)=-{\alpha \;\theta \;k}_{b}\frac{\Phi \left(X^{B},t\right)-\Phi \left(X^{A},t\right)}{||\xi ||}e\left[X^{A},X^{B}\right]-\left(\hat{\kappa }+\alpha^{2}{\theta \;k}_{b}\right)\frac{\Theta \left(X^{B},t\right)-\Theta \left(X^{A},t\right)}{||\xi ||}e\left[X^{A},X^{B}\right]$	$j\left(X^{A},t\right)=-{\;k}_{b}\frac{\Phi \left(X^{B},t\right)-\Phi \left(X^{A},t\right)}{||\xi ||}e\left[X^{A},X^{B}\right]-{\;k}_{b}\alpha \;\frac{\Theta \left(X^{B},t\right)-\Theta \left(X^{A},t\right)}{||\xi ||}e\left[X^{A},X^{B}\right]$

Where $\;\kappa_{E}$ is the electric conductivity tensor, κ is thermal conductivity tensor, Φ is electric potential and  α is Seebeck coefficient, Θ is the temperature, ρ is the mass density, $C_{v}$ is the specific heat capacity, q is the heat flux, j is the electrical charge flux, z the electrical charge of the carrier, and R is the charge source per unit of mass per unit time.

**Table 2 materials-13-02546-t002:** Material properties and dimensions.

Geometric Parameters	Material Properties
Length, L = 2 cm	Thermal conductivity κ=1.6 W/K.m
Width, W = 2 cm	${\mathrm{Heat}\;\mathrm{Capacity}\;C}_{v}{}_{\left(A\right)}=154.4\;J/K.\mathrm{kg}$
Thickness, t = 0.01 cm	${\mathrm{Density}\;\rho }_{\left(A\right)}=7740\;\mathrm{kg}/m^{3}$

**Table 3 materials-13-02546-t003:** Geometric dimensions and material properties [47] 2017, B. L. Wang.

Geometric Parameters	Material Properties
Length, L = 1.524 mm, Width, W = 1.4 mm,	$\;\alpha =1.849\times 10^{-4\;}v/K$ κ=1.701 W/K m

**Table 4 materials-13-02546-t004:** Comparison between the PD result and the results from references [46,47] in the case of constant material properties.

Horizontal Distance (mm)	Electric Potential (10^−3^ V) (This Study)	Electric Potential (10^−3^ V) Ref. [46]	Electric Potential (10^−3^ V) Ref. [47]	Temperature (K) (This Study)	Temperature (K) Ref. [46]	Temperature (K) Ref. [47]
0	4.603	4.622	4.622	273	273	273
0.762	2.311	2.311	2.311	285.45	285.3	285.3
1.143	1.143	1.156	1.156	291.8	291.8	291.7
1.524	0	0	0	298	298	298

**Table 5 materials-13-02546-t005:** Geometric Dimensions and Material Properties (temperature-dependent).

Geometric Parameters	Material Properties
Length (L) = 1.524 mm, Width (W) = 1.4 mm,	$\;\alpha =1.804\times 10^{-4}+3.598\times 10^{-7}\left(T-273\right)$ $\kappa =1.754-4.260\times 10^{-3}\left(T-273\right)$

where T is in Kelvin.

**Table 6 materials-13-02546-t006:** Comparison between peridynamic results and results from references [46,47] in the case of temperature-dependent material properties.

Horizontal Distance (mm)	Electric Potential (10^−3^ V) (This Study)	Electric Potential (10^−3^ V) Ref. [46]	Electric Potential (10^−3^ V) Ref. [47]	Temperature (K) (This Study)	Temperature (K) Ref. [46]	Temperature (K) Ref. [47]
0	4.35	4.636	4.622	273	273	273
0.381	3.362	3.524	3.514	279.17	279.1	279.1
0.762	2.309	2.380	2.375	285.28	285.3	285.3
1.143	1.19	1.205	1.205	291.56	291.6	291.6
1.524	0	0	0	298	298	298

**Table 7 materials-13-02546-t007:** Geometric Parameters and Material Properties.

Geometric Parameters	Material Properties
Length, L = 0.02 m Width, W = 0.02 m	Thermal conductivity κ=1.6 W/K.m
	${\mathrm{Heat}\;\mathrm{Capacity}\;C}_{v}{}_{\left(A\right)}=154.4\;J/K.\mathrm{kg}$
	${\mathrm{Density}\;\rho }_{\left(A\right)}=7740\;\mathrm{kg}/m^{3}$

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
