# Peer review of "A Peridynamic Computational Scheme for Thermoelectric Fields"

_materials, 2020, doi:10.3390/ma13112546_

Round 1
Reviewer 1 Report
This manuscript has good intent and can have good impact however the authors need to make the following changes before it can be published:
- The language of the paper needs to be revised, many grammatical errors need to be corrected.
- The literature review needs to be made more simple, concise and relevant to the manuscript at hand.
- Third and most important, the details of peridynamics theory and the coupling between fields, derivations of constants are largely missing. The authors state that this no other paper that discusses both thermal and electric fields in thermoelectric materials. In this case, it is imperative that they provide all the details of derivations so that others may find it useful. Citing other references is not enough.
Other specific comments are below:
- In the first line of the abstract, change ‘..heat and electric current’ to ‘heat flux and electric current’.
- Page 2, line 43 – “In general about two thirds of the energy generated is wasted in the form of heat in the most efficient heat engine.” Not sure that this is accurate, please provide a reference if so.
- Page 2, line 46 – change the word “pretty” or omit completely.
- Page 2, line 49 – change “reduce” to “reduces”
- Page 2, line 52 – what “interface” are the authors referring to?
- Page 2, line 52 – change “affect” to “affects”
- Page 2, line 53 – change “intern” to “in turn”
- Page 2, line 55 – change “results” to “results in”
- Page 2, line 68 – change “results” to “results in”
- Page 2, line 76 – rephrase “adequate accuracies”
- Page 2, line 93 – effective Poisson’s ratio in 3D and 2D plane strain is ¼ but for 2D plane stress is 1/3.
- Page 2, the authors seem to be missing many references in coupled peridynamics which they should include in their literature review. Hailong Chen and co-authors have a few papers on thermomechanical peridynamics implemented in MOOSE, Prakash and Seidel have a few papers on electromechanical and piezoresistivity in peridynamics, Diana and Carvelli have a recent paper on electromechanical micropolar peridynamics, Wildman and Gazonas have a paper on dielectric breakdown using peridynamics etc. Including these relevant references would be more useful to the reader than the extensive list of references on fracture, plasticity, composites etc. which are not directly related to the manuscript. If inclined to do so, only a few would be enough. It also appears that the authors have repeated many sentences and references in the first paragraph on page 2 and first paragraph on page 3. Therefore the reviewer suggests rewriting the literature review in a simple and concise manner with all the relevant references.
- Page 4, line 164, rewrite “here write” to “here we write”.
- Section 2.1.1 introduces peridynamic equations without any kind of formal introduction to peridynamics or thermoelectric effects! The authors MUST provide some basic information about peridynamics (integral equations, horizon, how the parameters like micromodulus are determined etc.). Including a figure would be even better. What do the authors mean by Fourier and Peltier effects, which direction are these couplings in? All of these details are a must for the reader to understand the manuscript.
- In equation (4), the microconductivities K hat and K_E hat are divided by the volume of the nodal point at X^A. Why ? Furthermore, the left hand side seems to be unchanged ?
- Derivations of 6a and 6b need to be provided.
Author Response
Response to the Reviewer #1
We really appreciate your careful reading of our manuscript and valuable suggestions. We have carefully considered the comments and have revised the manuscript accordingly. The comments and responses are summarized as follows:
Comments and Suggestions for Authors
- The language of the paper needs to be revised; many grammatical errors need to be corrected.
Response: we agree with this suggestion and tried our best to make the language more lucid by incorporating your detailed comments.
- The literature review needs to be made more simple, concise and relevant to the manuscript at hand.
Response: Thanks for the reviewer's suggestion; we include the following paragraph to make the literature review more exhaustive. We believe that general comments #2 and specific comment #12 can be merged and changed as follows:
Some other applications of PD for thermo mechanical and electromechanical coupling can be found in (34; 35; 36; 23; 37; 38; 39; 40; 41; 42). Chen et al (25) applied implicit PD formulation in the framework of MOOSE to study a coupled thermo-mechanical problems. In this article the authors used bond based PD (BB-PD) formulation with regular square discretization. Later the same authors (41) extended their study by reformulating the classical bond based PD (BB-PD) formulation to solve thermo-mechanical problem using irregular domain discretization. A more general and interesting formulation of ordinary state-based PD formulation to solve thermo-mechanical problems with irregular non uniform domain discretization can be found in (42). Zhang and Qiao (34) presented a new PD Scheme by extending the ordinary state-based peridynamic (OSB-PD) model to simulate damage initiation and propagation of bimaterials under thermo mechanical loading. Wang et al. (35) on the other hand investigated a coupled thermo mechanical BB-PD to simulate thermal shock cracking in rocks. The authors in this article also investigated the effect of inhomogeneous properties and the coefficients of thermal expansion on the cracking patterns. Very recently Bazazzadeh et al. (36) effectively developed a thermo mechanical PD model by exploiting the advantages of adaptive grid refinement to solve crack propagation problem in ceramic materials. Wildman and Gazonas (23) applied PD approach to study the failure of dielectric solids that are subjected to electric fields. In the peridynamic perspective of coupled electromechanical and electrical conduction models Prakash and Seidel (37) explored the beauty of PD to investigate the piezoresistive and electrical response of composite materials at nano scale by introducing electron hopping. Further Prakash and Seidel (38; 39) employed a coupled electromechanical PD framework to model the damage and deformation sensing capabilities of explosive materials without considering electron hopping. A recent work of Diana and Carvelli (40) implemented micropolar PD (MPPD) formulation to solve electromechanical problems by coupling the electrical conduction PD model with the mechanical micropolar formulation which removed Poisson’s ratio restrictions.
- Third and most important, the details of peridynamics theory and the coupling between fields, derivations of constants are largely missing. The authors state that this no other paper that discusses both thermal and electric fields in thermoelectric materials. In this case, it is imperative that they provide all the details of derivations so that others may find it useful. Citing other references is not enough.
Response: we appreciate the reviewer's suggestion, we rephrase the statement as follows “To the best of our knowledge, there are only few attempts by Migbar and his coworkers (24; 43) in the analysis of both the thermal and electric fields in thermoelectric plate. Apart from that we include the PD formulation of electrical conduction followed by PD formulation of thermal conduction to show some of the derivations and moved finally to the coupled thermoelectric phenomena.
- PD Electric conduction
In Electrical conduction phenomena, material points exchange electrical flux with material points inside its integration domain defined by the horizon as shown in Fig.1. With reference to Fig.1, consider the electrical conduction bond connecting material points and having volumes and respectively. The conductivity of electrical conduction bond is designated by. At this point we assume electric-current flows only along the length of the electrical conduction bond, material points interacts in a pair-wise manner, and also the bonds have electrical resistance of zero.
Figure 1, Peridynamic representation of problem domain; here Point is influenced by all points within its horizon is the horizon of δ is the radius of the horizon.
Based on the assumptions above, we can express the volumetric electrical flux of current passing through the bond in terms of the potential difference is obtained from Ohm’s law as follows:
(1)
Where:
is the unit vector along the bond, is Electric Potential, is the magnitude of the bond vector andis conductivity of the electrical bond. (Eq.1) represents the PD current flowing through a single electrical conduction-bond. Using the balance of charge for the bond, the time rate of change of total charge in the bond per unit bond volume must be equal to the net current flow through the bond per unit bond volume. In the presence of charge source term at material point the PD electrical conduction equation is given by:
(2)
is the volumetric charge source term. To obtain the conservation of charge equation for material point , due to current flow in all the bonds attached to point in its horizon, we integrate Eq. (2) over the horizon of .
(3)
Simplifying the left-hand side of Eq. (3), we have
(4)
Where: is the volume of the horizon of point . Similarly, the charge source term at point is obtained by the average of the same in all the bonds attached to in its horizon:
(5)
Therefore we may write the PD conservation of charge equation by equating eqs. (3-5) for any material point as:
(6)
Where: is the time rate of change of total charge, and in case of 2-D are defined as the microconductivity of the electrical bond.
In compact form eq. (6) is written as follows:
(7)
Where:
is the electric current flow density function, the time rate of change density, is electric potential difference among and is the applied current flux.
Borrowing the definition of PD heat flux from Florin and Monchai (27) and extend it to obtain the PD electric charge flux equation at any point we get:
(8)
Where is the particular area in the horizon of point with neighboring points of higher electrical potential than that at .
Other specific comments are below:
We really respect the reviewer's detailed specific comments. We make the changes as per the comment line by line.
- In the first line of the abstract, change ‘..heat and electric current’ to ‘heat flux and electric current’.
Response: Thermoelectric materials are materials that involve the coexistence of heat flux and electric current in the absence of magnetic field.
- Page 2, line 43 – “In general about two thirds of the energy generated is wasted in the form of heat in the most efficient heat engine.” Not sure that this is accurate, please provide a reference if so.
Response: we made the change as follows by incorporating its citation.
In general about 70% of the energy generated by gasoline engine is wasted in the form of heat (2).
- Page 2, line 46 – change the word “pretty” or omit completely.
Response: Deleted
- Page 2, line 49 – change “reduce” to “reduces”
Response: Changed as per the suggestion
- Page 2, line 52 – what “interface” are the authors referring to?
Response: Corrected as “cracks with in the material”
- Page 2, line 52 – change “affect” to “affects”
Response: Changed as per the suggestion
- Page 2, line 53 – change “intern” to “in turn”
Response: Changed as per the suggestion
- Page 2, line 55 – change “results” to “results in”
Response: Changed as per the suggestion
- Page 2, line 68 – change “results” to “results in”
Response: Changed as per the suggestion
- Page 2, line 76 – rephrase “adequate accuracies”
Response: Rephrased as good accuracies
- Page 2, line 93 – effective Poisson’s ratio in 3D and 2D plane strain is ¼ but for 2D plane stress is 1/3.
Response: Changed as per the suggestion
- Page 2, the authors seem to be missing many references in coupled peridynamics which they should include in their literature review. Hailong Chen and co-authors have a few papers on thermomechanical peridynamics implemented in MOOSE, Prakash and Seidel have a few papers on electromechanical and piezoresistivity in peridynamics, Diana and Carvelli have a recent paper on electromechanical micropolar peridynamics, Wildman and Gazonas have a paper on dielectric breakdown using peridynamics etc. Including these relevant references would be more useful to the reader than the extensive list of references on fracture, plasticity, composites etc. which are not directly related to the manuscript. If inclined to do so, only a few would be enough. It also appears that the authors have repeated many sentences and references in the first paragraph on page 2 and first paragraph on page 3. Therefore the reviewer suggests rewriting the literature review in a simple and concise manner with all the relevant references.
Response: Repeated sentences and some of irrelevant citations have been carefully removed. More relevant citations have been incorporated as per the reviewer’s suggestion as explained in #2 above.
- Page 4, line 164, rewrite “here write” to “here we write”.
Response: Changed as per the suggestion
- Section 2.1.1 introduces peridynamic equations without any kind of formal introduction to peridynamics or thermoelectric effects! The authors MUST provide some basic information about peridynamics (integral equations, horizon, how the parameters like micromodulus are determined etc.). Including a figure would be even better. What do the authors mean by Fourier and Peltier effects, which direction are these couplings in? All of these details are a must for the reader to understand the manuscript.
Response: We include a single paragraph of a brief introduction on thermoelectric effects. Regarding peridynamic equations and PD terminologies we discussed it by including one additional figure. (Fig.1).
Over utilization of fossil fuel, the world’s energy demand and climate change are the main factors that push forward the research in the area of new energy materials. The main criteria of these materials are mainly sustainability and environmental friendliness. In this context, thermoelectric materials (TEMs) are excellent candidates due to their simplicity, compactness, scalability, sustainability, portability, reliability and environmental friendliness. Moreover, TEMs are quite capable in converting temperature change to electric voltage or electricity to heat. The first scenario couples electric field and temperature gradient and it is known as Seebeck effect while the second scenario is known as Peltier effect which is the reverse of the first one. Seebeck effect was discovered by Seebeck in 1821 and Peltier effects by Peltier in 1855, these two effects commonly called thermoelectric effect (44). Therefore we dedicate this section to the development of peridynamic governing equations for thermoelectricity. These equations are described as constitutive equations and the conservation laws of thermoelectricity. The detailed description and derivations of these equations using classical approach may be found in (45).
- In equation (4), the microconductivities K hat and K_E hat are divided by the volume of the nodal point at X^A. Why? Furthermore, the left hand side seems to be unchanged ?
Response: Because microconductivity has the units of conductivity per unit volume. When we multiply it by volume we get bond conductivity. Regarding the LHS of Eq. (3) and Eq. (4); yes they are the same the only difference is Eq. (3) represents the law of conservation of energy for system of thermoelectric without the heat source term whereas Eq. (4) takes in to account the source term.
- Derivations of 6a and 6b need to be provided.
Response: we actually borrowed those equations from S. Oterkus et al. (2014) for detailed derivation of microconductivity readers are advised to visit Appendix B. of this article.

Reviewer 2 Report
The evaluated paper is interesting. The study presents a Peridynamic (PD) computational scheme for thermoelectric fields. The PD described theory allows considering discontinuities in thermal and electric fields due to the cracks in materials. Unlike classical mathematic theories, this method does not require any criterium to represent and predict the behavior of discontinuities. This article presents an interesting approach to study the thermal and electrical transport processes of thermoelectric materials. To validate this peridynamic technique, numerous examples showed that the results were in good agreement with those from literature. The literature is very rich and cited correctly. This paper will certainly interest research in the field of thermoelectricity. The content is very clear and well-organized. The language is clear. The paper is recommended for publication without correction.
Author Response
Response to the Reviewer #2
We really appreciate your careful reading of our manuscript and valuable suggestions.
Reviewer 3 Report
line 164, 165) “Fourier and Peltier effects”, „The presence of Fourier effect is due to temperature gradient” you mean Seebeck effect?
line 168) not all variables are explained here
line 254) “parameters are specified in Table 2” – Table 2 is missing. There is Tab. 1, next one is Tab. 3.However, in my opinion the phrase should be “parameters are specified in Table 1”. Numbers of Tab. 3 and following should be ‘decremented’.
The legend in Fig. 3a suggests, there is one characteristic only (but the figure subtitle tells about PD and analytical characteristics). The characteristic is missing or the legend should be improved.
Author Response
Response to the Reviewer #3
We are grateful for your comments and valuable suggestions. We have carefully considered the comments and have revised the manuscript accordingly. The comments and responses are summarized as follows:
Comments and Suggestions for Authors
- line 164, 165) “Fourier and Peltier effects”, „The presence of Fourier effect is due to temperature gradient” you mean Seebeck effect?
Response: we really appreciate the reviewer’s suggestion. The change has been made as follows:
The presence of Seebeck effect is due to temperature gradient and that of Peltier effects is due to the change in an electric current.
- line 168) not all variables are explained here
Response: We included all the variables as per the suggestion.
- line 254) “parameters are specified in Table 2” – Table 2 is missing. There is Tab. 1, next one is Tab. 3.However, in my opinion the phrase should be “parameters are specified in Table 1”. Numbers of Tab. 3 and following should be ‘decremented’.
Response: We agree with this suggestion and rectified the confusion.
- The legend in Fig. 3a suggests, there is one characteristic only (but the figure subtitle tells about PD and analytical characteristics). The characteristic is missing or the legend should be improved.
Response: It is typing error it is corrected as 2-D temperature variations from PD (Symmetric boundary).

Reviewer 4 Report
Assefa, Lai and Liu present a perydynamic study for thermoelectricity. This is the first time I hear about "perydynamics". In any event, I didn't quite get an idea about what this scheme might be. Looking at the equations it looks like simply replacing common spatial derivatives by a division with this mysterious and undefined ||ksi||.
The manuscript is written awfully. If it ever is to see the light of day, very heavy copy editing is required. It is replete with misspellings, even within Figures, strange word orders, missing words etc.
At line 123 I started to feel like Bill Murray in Groundhog Day... Quite troubling is that the English quality of this copy-pasted section from lines 90-122 is dramatically better than the rest of the paper.
As the description of the model starts in Eqs (10 and (2) many symbols are undefined although perhaps they get defined after line 200?
As I got to the numerical section, looking at the material parameters of Table 1, I nearly quit reading further. What sense does it make to use such a tremendously large thermal conductivity for thermoelectric materials? What superdense material would have such enormous density? Is this a study of the core of the Earth? And what material has such tremendous heat capacity?
Author Response
Response to the Reviewer #4
We are appreciating for your comments and valuable suggestions. We have carefully considered the comments and have revised the manuscript accordingly. The comments and responses are summarized as follows:
Comments and Suggestions for Authors
- Assefa, Lai and Liu present a perydynamic study for thermoelectricity. This is the first time I hear about "perydynamics". In any event, I didn't quite get an idea about what this scheme might be. Looking at the equations it looks like simply replacing common spatial derivatives by a division with this mysterious and undefined ||ksi||.
Response: we respect the reviewer’s suggestion. We tried to include the basic and fundamental topic on peridynamics in section 2.1. Further we defined each variable at the moment they appear.
- The manuscript is written awfully. If it ever is to see the light of day, very heavy copy editing is required. It is replete with misspellings, even within Figures, strange word orders, missing words etc.
Response: we tried our best to review the language and rectify the misspellings.
- At line 123 I started to feel like Bill Murray in Groundhog Day... Quite troubling is that the English quality of this copy-pasted section from lines 90-122 is dramatically better than the rest of the paper. As the description of the model starts in Eqs (1) and (2) many symbols are undefined although perhaps they get defined after line 200?
Response: we defined each and every variable at their first appearance.
- As I got to the numerical section, looking at the material parameters of Table 1, I nearly quit reading further. What sense does it make to use such a tremendously large thermal conductivity for thermoelectric materials? What superdense material would have such enormous density? Is this a study of the core of the Earth? And what material has such tremendous heat capacity?
Response: We agree with this suggestion, actually the material properties are similar with those in Table 7, Table 1 mixed up while transferring files. Actually it should be Table 2 because Table 1 is: summery of comparison between PD and Classical Equations (Local Vs. PD formulation for TEM) which appeared before this one. Therefore we corrected it as follows.

Round 2
Reviewer 1 Report
No further comments to the authors.
Author Response
We really appreciate the reviewer for reading and commenting on our article.
Reviewer 4 Report
The manuscript has been considerably improved, now I can get a sense of what this PD simulation might be doing. Many self-justifying examples have been included that can be compared with either simple analytical solutions, or with results from more well established simulations by finite elements.
However, the paper suddenly stops there. Where are the novel examples. of geometries with cracks that would be difficult to analyse by FEM?
As it is, the paper seems like only an introduction.
Author Response
Response to Reviewer 4:
We really appreciate the reviewer’s valuable suggestion. We have carefully considered the comments incorporate one more numerical example to show the novelty of our PD model and summarized as follows:
Example 4: Heat conduction with a moving crack
To substantiate the novelty of our PD model, we provide an example that demonstrates the beauty of PD in treating propagating cracks. The dimensions, number of material points and material properties are considered to be the same as the one in example 3 except the crack in this example is center crack of length 2a. Here we assume an initial crack of size of starting from point A (X= -0.5cm) as shown in Fig.17 below. The bottom and top edges of the plate are exposed to a temperature of and respectively while the right and left edges kept insulated.
The crack initiates with a length of and expected to travel along the X direction as in Fig. 17. Here the crack travels at the velocity of 0.0125cm/s as it approaches point B(X=0.5cm) and hence ensuing length of the crack is measured to be 1cm.
Figure 17. Geometry and the boundary conditions of example 4
Fig. 18 below shows our PD results in the case of moving cracks. In this particular example, it is not possible to compare our PD results with that of FEM due to the fact that FEM requires new models for each and every steps of simulation and hence it is time consuming and cumbersome. Thus it is inspiring to conclude at this point that peridynamic formulation is one of the promising computational tools in handling growing crack.
|
(a)Temperature contour (°C) at t = 10 s |
(b)Temperature contour (°C)at t = 20 s |
|
(c)Temperature contour (°C)at t = 40 s |
(d)Temperature contour (°C)at t = 50 s |
|
(e)Temperature contour (°C)at t = 60 s |
(f)Temperature contour (°C)at t = 60 s |
Figure 18. Temperature contour of plate with crack
